# Prognostic Factors in Patients Treated with Pembrolizumab as a Second-Line Treatment for Advanced Biliary Tract Cancer

**DOI:** 10.3390/cancers14174323

**Published:** 2022-09-03

**Authors:** Chan Su Park, Min Je Sung, So Jeong Kim, Jung Hyun Jo, Hee Seung Lee, Moon Jae Chung, Seungmin Bang, Seung Woo Park, Si Young Song, Jeong Youp Park

**Affiliations:** 1Division of Gastroenterology, Department of Internal Medicine, Yonsei Institute of Gastroenterology, Yonsei University College of Medicine, 50 Yonsei-ro, Seodaemun-gu, Seoul 03722, Korea; 2Department of Gastroenterology, CHA Bundang Medical Center, CHA University, 59 Yatap-ro, Bundang-gu, Seongnam-si 13496, Korea

**Keywords:** prognostic factor, pembrolizumab, advanced biliary tract cancer

## Abstract

**Simple Summary:**

The five-year survival rate for advanced biliary tract cancer (BTC) is still less than 10%. Moreover, there is no established second-line chemotherapy after the failure of the first-line chemotherapy. Pembrolizumab is known to be an effective treatment in advanced BTC, but the prognostic factors of pembrolizumab are unknown. The purpose of this study is to provide guidelines for selecting treatment alternatives for BTC patients with refractory to gemcitabine-based chemotherapy, by exploring the prognostic factors of pembrolizumab in BTC.

**Abstract:**

Some BTC types respond to pembrolizumab, but there are no known prognostic factors to predict its treatment benefits. In this study, we attempted to identify the prognostic factors associated with pembrolizumab as a second-line treatment for gemcitabine-refractory BTC. This retrospective and single tertiary-center study involved all the consecutive patients (n = 80) with refractory advanced BTC, who were diagnosed as programmed cell death ligand 1-positive and treated with pembrolizumab between August 2017 and February 2021. The overall survival (OS) was analyzed using Cox regression analysis. The median OS was 6.0 months [95% confidence interval (CI): 3.87–8.20]; median progression-free survival was 1.9 months (95% CI: 1.82–1.98); and the response rate was 15.9%. In the multivariate Cox regression analysis, the TB [adjusted hazard ratio (HR) = 2.286; 95% CI: 1.177–4.440; *p* = 0.015), albumin levels (adjusted HR = 0.392; 95% CI: 0.211–0.725; *p* = 0.003), ALP levels (adjusted HR = 1.938; 95% CI: 1.105–3.400; *p* = 0.021), and LMR (adjusted HR = 0.325; 95% CI: 0.173–0.609; *p* < 0.001) were identified as significant variables associated with the OS. High albumin levels and LMR and low ALP levels and TB were significantly associated with better OS in patients treated with pembrolizumab.

## 1. Introduction

In recent years, the global incidence of biliary tract cancers (BTCs), including intrahepatic, extrahepatic, and gall bladder cancers, has been increasing worldwide. However, the prognosis of BTC is poor, with the median overall survival (OS) typically not extending beyond one year [1,2,3]. Although surgical resection is important for curative treatment, the disease is often advanced at the time of diagnosis, due to the tumor location and asymptomatic nature of the disease. Palliative chemotherapy is the only treatment option for inoperable patients [4,5,6].

According to the latest National Comprehensive Cancer Network guidelines, cisplatin–gemcitabine combined chemotherapy is recommended as the first-line treatment for inoperative patients [7]. However, there is no standard second-line treatment for patients who fail the first-line gemcitabine-based chemotherapy. Therefore, it is important to find an effective second-line drug or regimen to increase the survival rate of patients with inoperable BTC [8].

Pembrolizumab, an anti-programmed cell death (PD)-1 monoclonal antibody, is an anticancer agent that has shown substantial benefits in various solid tumors and is approved as a standard therapeutic drug for several cancers, including lung cancer and melanoma [9,10,11,12]. Considering that programmed cell death ligand-1 (PD-L1) expression is reported in 9.1–72.2% of patients with BTC [13,14,15], anti-PD-1 or PD-L1 agents may be promising treatment options for some patients with BTC. A few studies have demonstrated the efficacy of pembrolizumab in some patients with PD-L1-positive advanced BTC [16,17]. However, there are no known clinical prognostic factors for predicting the treatment effects.

In this study, we attempted to identify the prognostic factors associated with pembrolizumab as a second-line treatment for gemcitabine-refractory BTC to determine the efficacy of pembrolizumab over other treatment options.

## 2. Materials and Methods

### 2.1. Patients

From August 2017 to February 2021, 142 patients who received pembrolizumab as a second-line or higher treatment were included in this single-center study with the following inclusion criteria: (1) ≥20 years of age; (2) locally advanced or metastatic BTC, histologically or cytologically confirmed to be inoperable; (3) radiologically confirmed presence of progressive disease or intolerance to first-line chemotherapy; (4) ≥1% PD-L1-positive tumor cells as assessed using immunohistochemical (IHC) staining; and (5) at least one cycle of pembrolizumab treatment.

Sixty-two patients were excluded according to the following criteria: (1) inability to assess tumor response; (2) no gemcitabine chemotherapy before pembrolizumab treatment, and (3) no PD-L1 IHC staining examination. Finally, 80 patients were enrolled in this study.

This study was approved by the Ethics Committee of Severance Hospital, Seoul, Korea, and conducted in accordance with the Declaration of Helsinki (Institutional Review Board number: 2022-1877-001).

### 2.2. Histopathological Analysis

PD-L1 expression was assessed by IHC staining, using SP263 (Ventana Benchmark Ultra, Tuscon, AZ, USA), the 22C3 pharmDx kit (Agilent Technologies, Santa Clara, CA, USA), or E1L3N XP Rabbit mAb (Cell Signaling Technology, Danvers, MA, USA). If more than 1% of viable tumor cells had PD-L1, they were considered PD-L1-positive [18]. The tumor proportion score (TPS) was defined as the percentage of viable tumor cells showing partial or complete membrane-staining at any intensity [19]. The combined positive score (CPS) was defined as the number of PD-L1-positive cells (tumor cells, lymphocytes, and macrophages), divided by the total number of viable tumor cells, and multiplied by 100 [20].

The microsatellite stability was assessed using antibodies specific for mismatch repair proteins, including mutL homolog 1 (1:10; clone G168-15; BD Pharmingen, San Jose, CA, USA), mutS homolog (MSH)-2 (1:100; clone FE11; Calbiochem, San Diego, CA, USA), MSH6 (1:100; clone EP49; Novus Biologicals, Centennial, CO, USA), and PMS1 homolog 2 (1:50; clone A16-4; BD Pharmingen).

### 2.3. Treatment and Assessment

The patients received an intravenous infusion of pembrolizumab (200 mg) every three weeks for two years, or until the disease progression or life-threatening adverse events (AEs) were identified. If patients were intolerable due to a serious or poor condition, we considered a first-dose reduction or an administration delay to continue treatment. To evaluate the treatment efficacy, we routinely performed response evaluation every three cycles using abdominal- or chest-computed tomography, or both, according to RECIST version 1.1.

### 2.4. Outcomes

The primary efficacy endpoint was the overall survival (OS), defined as the period between the first pembrolizumab administration as the baseline and the event or censoring. The date of occurrence of the event was the date of death due to all causes, and the date of censoring was the date on which it was last judged that no incident had occurred. If patient follow-up was not possible, the OS was calculated as the last date of survival.

The secondary endpoints were AEs, progression-free survival (PFS), and the response rate. The AEs were undesirable and unintended signs, symptoms, or diseases that occurred in a subject after the first dose of pembrolizumab, and the dose was not necessarily causal with pembrolizumab.

### 2.5. Assessment of Treatment-Related AEs

To monitor treatment-related AEs, doctors and nurses carefully assessed the incidence of AEs at each visit. The category and severity grade of the AEs were accurately noted in the patient’s medical records. The treatment-related AEs were assessed according to the National Cancer Institute Common Toxicity Criteria, version 4.0. Any treatment delays or discontinuations associated with the AEs were documented along with the reasons.

### 2.6. Statistical Analysis

The data are expressed as the mean ± standard deviation, median (range), or n (%). The survival analysis for the OS and PFS was performed using the Kaplan–Meier method and compared using the log-rank test. To identify prognostic factors related to OS, we performed a multivariate Cox proportional hazard regression analysis using the significant variables in the univariate analysis; forward selection was used to eliminate the non-significant variables. The statistical significance was set at a two-tailed *p*-value < 0.05. The statistical analyses were performed using SPSS version 26.0 (PASW Statistics Inc., Chicago, IL, USA). The cutoff values of the variables were set by referring to previous articles.

## 3. Results

### 3.1. Baseline Characteristics

The baseline characteristics of the 80 patients enrolled in this study from August 2017 to February 2022 are summarized in Table 1. The median age was 66.4 (range: 43–85) years, and 51 (63.7%) patients were male. The primary tumor locations were intrahepatic in 55 patients, extrahepatic in 21 patients, and the gall bladder in 4 patients. Staging revealed that 10 patients had locally advanced disease, while 70 patients had metastatic disease. The median number of cycles of pembrolizumab was three (range: 1–30), and disease progression was the most common cause of discontinuation of administration (n = 52, 65%).

Among the 142 patients treated with pembrolizumab for BTC, 80 were enrolled in this study by the inclusion and exclusion criteria (Figure 1).

### 3.2. Treatment and Clinical Outcomes

The clinical outcomes of pembrolizumab are summarized in Table 2. The median OS of the patients treated with pembrolizumab was 6.0 months (range: 0.3–44.1) (Figure 2). Moreover, 1 (1.25%) patient had complete remission; 5 (6.25%) had partial remission; 26 (32.5%) had stable disease; and 43 (53.75%) had progressive disease. The median PFS of pembrolizumab was 1.9 months (range: 0.3–43.4) (Figure 3).

The results of the univariate and multivariate Cox analysis to identify the factors affecting the OS of patients treated with pembrolizumab are summarized in Table 3. The OS was significantly correlated with the tumor burden (TB; *p* = 0.010), albumin levels (*p* < 0.001), alkaline phosphatase (ALP) levels (*p* = 0.066), lymphocyte-to-monocyte ratio (LMR; *p* = 0.003), neutrophil-to-lymphocyte ratio (*p* < 0.001), platelet-to-lymphocyte ratio (PLR; *p* < 0.001), and carbohydrate antigen (CA)-19-9 levels (*p* = 0.022). The multivariate Cox analysis was performed by collecting variables that showed statistically significant results in the univariate Cox analysis (Table 3). The TB base (*p* = 0.015), albumin levels (*p* = 0.003), ALP levels (*p* = 0.021), and LMR (*p* < 0.001) were found to be significantly associated with the OS, and we estimate the overall survival of the entire study population (n = 80) according to the LMR (cut off value = 2.5) using the Kaplan–Meier method (Figure 4).

### 3.3. Treatment-Related AEs

Most of the AEs were mild-to-moderate in severity. Thirty-two (40%) patients experienced side effects related to pembrolizumab administration (Table 4). Among them, four patients, including three patients with an acute kidney injury and one patient with immune-related pneumonitis, stopped chemotherapy due to grade 3/4 side effects. All other patients complained of mild, grade 1/2 side effects. Of the grade 1/2 side effects, general weakness was the most common in eight patients (10.0%), followed by hypothyroidism and an itching sensation in three patients (3.8%). None of the patients underwent a dose reduction due to the side effects, and there were no cases of treatment-related deaths.

## 4. Discussion

Immunotherapy has emerged as a new treatment option for various types of cancer, including advanced-stage BTC. In particular, pembrolizumab treatment for PD-L1-positive BTC shows promising clinical results in several studies. However, there are no cost-effective or less invasive prognostic markers available for patients with advanced BTC treated with pembrolizumab. In this study, we found that high albumin levels and LMR, and low ALP levels and TB, were significantly associated with better OS in patients treated with pembrolizumab.

Our study showed that the median PFS and OS were 1.9 and 6.0 months, respectively, in patients with BTC, similar to the results of other prospective clinical trials using pembrolizumab in pretreated patients with advanced BTC. In the KEYNOTE-028 study of 24 PD-L1-positive patients with progressive BTC, the median PFS and OS were 1.8 and 5.7 months, respectively [2]. In the KEYNOTE-158 study of 104 patients with a PD-L1 positivity rate of 58.7%, the median PFS and OS were 2.0 and 7.4 months, respectively [2]. A univariate Cox analysis was performed on 13 variables of interest to identify factors affecting the OS, and statistically significant correlations were found in 7 variables (TB base, albumin levels, ALP levels, LMR, NLR, PLR, and CA19-9). Meanwhile, the *p*-value of ALP was 0.066, which was below the significance level of 5% but was allowed under clinical judgment. Importantly, all three variables related to lymphocytes correlated with the OS. We found that high lymphocyte levels were associated with better OS. The NLR and LMR are well-known systemic inflammatory response markers and are important prognostic factors in several other cancers treated with pembrolizumab. Current evidence suggests that cell-mediated cytotoxicity can be attenuated if the number of effector T cells is insufficient. After tumor infiltration, the circulating monocytes can differentiate into macrophages and contribute to both tumor growth and reduced immune surveillance, in response to a wide range of chemokines and growth or differentiation factors [21]. The tumor-associated macrophages are also involved in accelerating angiogenesis, invasion, migration, and metastasis, and suppressing the body’s autoimmune response to tumor cells [22]. Thus, the prognostic effect of LMR in patients with BTC may be related to the tumor-infiltrating immune cells, such as tumor-infiltrating lymphocytes or tumor-associated macrophages.

Multivariate analysis showed that the OS of gemcitabine-refractory PD-L1-positive patients with BTC treated with pembrolizumab was significantly related to the TB, albumin levels, ALP levels, and LMR. Contrary to expectations, no relationship was observed between the PD-L1 expression and OS in this study. Previous studies aimed to demonstrate the relationship between PD-L1 positivity and pembrolizumab efficacy; however, satisfactory results were not obtained [16]. Accordingly, there is an urgent need for additional research on new molecular markers that can predict the efficacy of pembrolizumab in patients with advanced BTC. Further studies are needed on peripheral blood cell biomarkers, such as the circulating tumor DNA, peripheral immune cells, peripheral cytokines, peripheral blood T-cell receptors, lactate dehydrogenase, interleukin-8, and soluble PD-L1, to determine the immune checkpoint inhibitor efficacy [23,24]. Many studies report that the biliary tract originates from the primitive gut and has a very similar morphology, particularly in the extrahepatic tract, and many researchers have demonstrated that DCLK1 (Doublecortin-like kinase 1) is a cancer stem cell marker in the intestinal tum [25,26,27]. Based on this background, further analysis of DCLK1 expression, before and after pembrolizumab treatment, is needed.

Most AEs related to pembrolizumab in the 32 patients were mild, with general weakness being the most common. Patients who experienced grade 3/4 side effects had to discontinue chemotherapy because of an acute kidney injury. During pembrolizumab administration, creatinine levels were slightly elevated, which worsened due to septic or hypovolemic shock accompanied by poor oral intake; dialysis was performed in such patients.

The limitation of this study is that patients who were in poor condition due to long-term chemotherapy received only one or two cycles of pembrolizumab before progression was confirmed. Hence, the efficacy and safety of pembrolizumab may not have been properly reflected in these patients. The heterogeneous patient group was another limitation of this study, as pembrolizumab had been used as a second-, third-, or higher-line treatment in different cases. Due to the limitation of the small cohort analysis, further studies are needed as more patient groups are gathered.

## 5. Conclusions

In conclusion, we identified the prognostic factors related to the OS of patients with advanced BTC treated with pembrolizumab. In these patients, the OS tended to increase with an increase in albumin levels and LMR, and a decrease in TB base and ALP levels. Therefore, this study provides evidence for predicting the prognosis of PD-L1-positive patients with advanced gemcitabine-refractory BTC using simple clinical factors. Further studies with more patients are needed to validate these prognostic factors to predict who would benefit most from pembrolizumab treatment for advanced BTC.

## Figures and Tables

**Figure 1 cancers-14-04323-f001:**
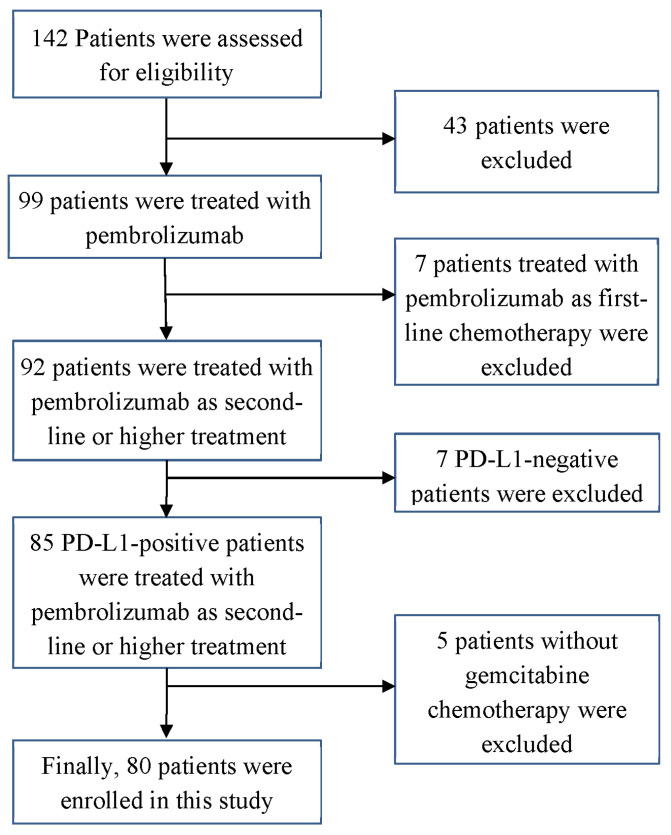
Patient flowchart. A total of 142 patients were screened, and, finally, 80 were enrolled in this study based on the inclusion and exclusion criteria for patients. PD-L1, programmed cell death-ligand 1.

**Figure 2 cancers-14-04323-f002:**
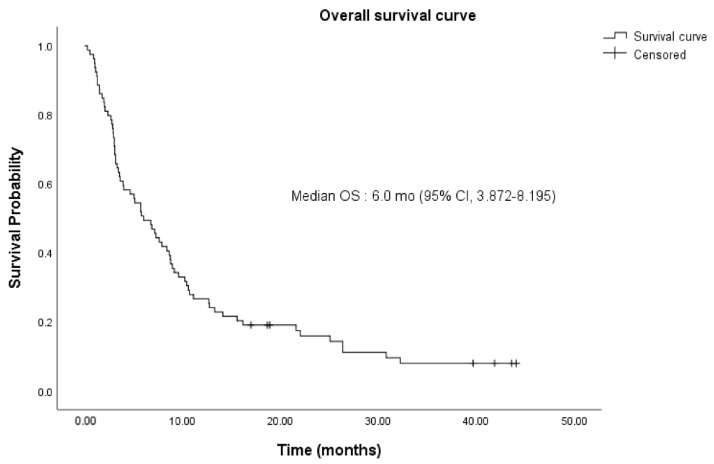
Kaplan–Meier estimates of the overall survival (OS) of the entire study population (n = 80). Median OS was 6.0 (95% confidence interval (CI), 3.872–8.195) months. OS, overall survival; mo, month; CI, confidence interval.

**Figure 3 cancers-14-04323-f003:**
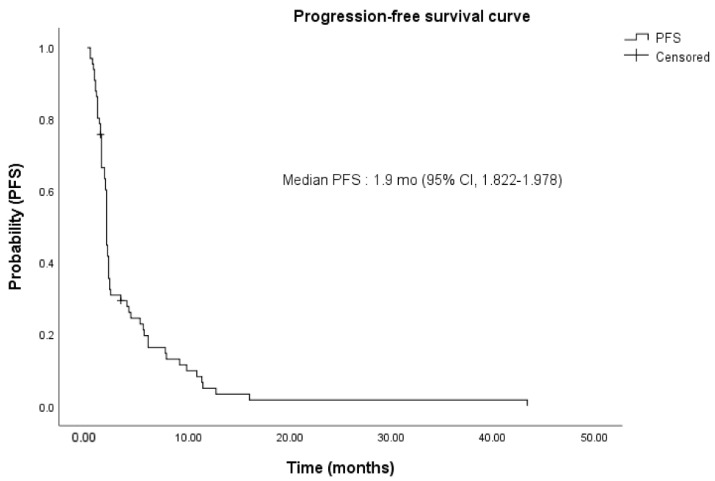
Kaplan–Meier estimates of the progression-free survival (PFS) of the entire study population (n = 80). Median PFS was 1.9 (95% CI, 1.822–1.978) months. PFS, progression-free survival; mo, month; CI, confidence interval.

**Figure 4 cancers-14-04323-f004:**
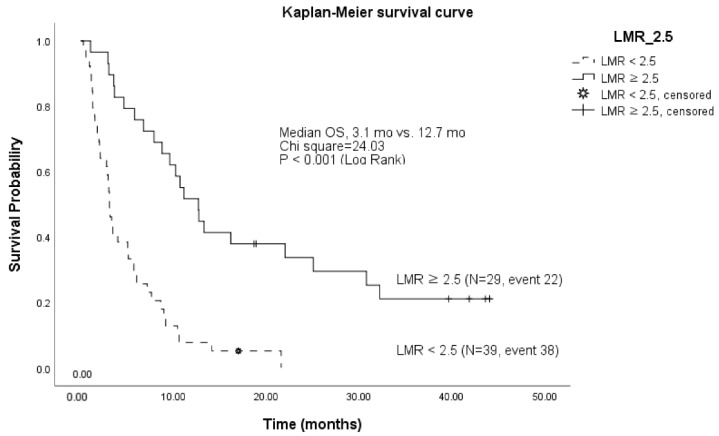
Kaplan–Meier survival curve of lymphocyte-to-monocyte ratio (LMR). Cut-off = 2.5. LMR, lymphocyte-to-monocyte ratio; mo, months.

**Table 1 cancers-14-04323-t001:** Baseline characteristics of patients enrolled in this study.

Characteristics	No. (%) (Total n = 80)
Age, average (range, year)	66.4 (43–85)
<65	29 (36.3%)
≥65	51 (63.7%)
Sex	
Male	51 (63.7%)
Female	29 (36.3%)
Performance status	
ECOG 0/1/2	60/18/2
Primary tumor location	
Intrahepatic/extrahepatic/gallbladder	55/21/4
Disease setting at presentation	
Locally advanced/metastatic	10/70
Site of metastasis	
Liver	49 (61.3%)
Intra-abdominal lymph node	40 (50.0%)
Peritoneum	23 (28.8%)
Bone	6 (7.5%)
Lung	8 (10.0%)
Histological grading	
Well/moderate/poorly/unknown	5/38/26/11
PD-L1 positive (≥1%)	
1–5/6–49/≥50 (TPS)	43/13/11
1–5/6–49/≥50 (CPS)	13/18/11
MSI-High	15 (18.8%)
Laboratory results	
Hemoglobin (g/dL)	10.2 ± 1.9
Albumin (g/dL)	3.8 ± 0.6
AST (IU/L)	30.8 ± 16.0
ALT (IU/L)	37.0 ± 68.7
ALP (IU/L)	204.0 ± 219.9
Total bilirubin (mg/dL)	0.8 ± 0.6
CA 19-9 (U/mL)	1755.0 ± 3609.2
Clinical information	
Number of prior therapies (1/2/≥3)	61/12/7
Cycles of pembrolizumab	3.0 (1–30)
≤3/>3	44 (55.0%)/36 (45.0%)
Combination treatment	1
Follow-up duration (months)	6.4 (0.3–44.1)

ECOG, Eastern Cooperative Oncology Group; PD-L1, programmed cell death-ligand 1; MSI-High, high microsatellite instability; AST, aspartate aminotransferase; ALT, alanine aminotransferase; ALP, alkaline phosphatase; CA 19-9, carbohydrate antigen 19-9; combination treatment, pembrolizumab plus radiation therapy.

**Table 2 cancers-14-04323-t002:** Clinical outcomes of patients.

Variable	RECIST v1.1 (n = 80)
Objective response	
CR	1 (1.25%)
PR	5 (6.25%)
SD	26 (32.5%)
PD	43 (53.75%)
Not evaluable	5 (6.25%)
ORR (CR + PR) (%)	6 (7.5%)
ORR by primary tumor site	
Intrahepatic cholangiocarcinoma	6/56 (10.7%)
Extrahepatic cholangiocarcinoma	0/21(0%)
Gallbladder cancer	0/4 (0%)
DCR (SD + CR + PR) (%)	32 (40%)
DCR by primary tumor site	
Intrahepatic cholangiocarcinoma	26/56 (46.4%)
Extrahepatic cholangiocarcinoma	5/21 (23.8%)
Gallbladder cancer	1/4 (25%)
Progression-free survival (95% CI, month)	1.9 (0.3–43.4)
Overall survival (95% CI, month)	6.0 (0.3–44.1)

CR, complete remission; PR, partial response; SD, stable disease; PD, progressive disease; ORR, overall response rate; DCR, disease control rate; CI, confidence interval.

**Table 3 cancers-14-04323-t003:** Cox analysis results.

Variable (Cut-Off)	Univariate		Multivariate	
	*p*-Value	HR (95% CI)	*p*-Value	HR (95% CI)
TPS	0.995	1.000 (0.992–1.008)		
CPS	0.237	0.994 (0.985–1.004)		
TB base (10 mm)	0.010	1.101 (1.023–1.184)	0.015	2.286 (1.177–4.440)
MSI-H	0.162	1.652 (0.817–3.339)		
Albumin (3.5 g/dL)	<0.001	0.346 (0.216–0.553)	0.003	0.392 (0.211–0.725)
ALP (151 IU/L)	0.066	1.001 (1.000–1.002)	0.021	1.938 (1.105–3.400)
AST	0.108	0.987 (0.970–1.003)		
ALT	0.859	1.000 (0.997–1.004)		
LMR (2.5)	0.003	0.721 (0.579–0.896)	<0.001	0.325 (0.173–0.609)
NLR	<0.001	1.277 (1.177–1.387)		
PLR	<0.001	1.005 (1.002–1.007)		
CA19-9	0.022	1.000 (1.000–1.000)		
BMI	0.687	0.985 (0.915–1.060)		

TPS, tumor proportion score; CPS, combined positive score; TB, tumor burden; MSI-H, high microsatellite instability; AST, aspartate aminotransferase; ALT, alanine aminotransferase; LMR, lymphocyte-to-monocyte ratio; NLR, neutrophil-to-lymphocyte ratio; PLR, platelet-to-lymphocyte ratio; CA 19-9, carbohydrate antigen 19-9; BMI, body mass index.

**Table 4 cancers-14-04323-t004:** Pembrolizumab-related adverse events.

Adverse Events	Grade 1/2 (n = 30)	Grade 3/4 (n = 2)
	No. (%)	No. (%)
General weakness	8 (10.0%)	0
Anemia	1 (1.3%)	0
Acute kidney injury	2 (2.5%)	2 (2.5%)
Diarrhea	2 (2.5%)	0
Facial edema	1 (1.3%)	0
Fatigue	2 (2.5%)	0
Hypothyroidism	3 (3.8%)	0
Skin rash	2 (2.5%)	0
Itching sensation	3 (3.8%)	0
Thrombocytopenia	1 (1.3%)	0
Peripheral neuropathy	2 (2.5%)	0
AST/ALT elevation	1 (1.3%)	0
Drug-induced pneumonitis	2 (2.5%)	0

AST, aspartate aminotransferase; ALT, alanine aminotransferase; No, number.

## Data Availability

The data presented in this study are available in this article. Data sharing not applicable.

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
