# Peer review of "Prognostic Factors in Patients Treated with Pembrolizumab as a Second-Line Treatment for Advanced Biliary Tract Cancer"

_cancers, 2022, doi:10.3390/cancers14174323_

Round 1
Reviewer 1 Report
Park et al. described effects of pembrolizumab on BTC, in particular PD-L1. Unfortunately, they used a very small cohort of patients, however, the manuscript was well written and very interesting.
In my opinion, authors should add very few points in their work.
In fact, as demonstrated in many papers the biliary tree as origin by primitive gut, and show a very similar morphology, especially in the extra-hepatic tract. Many researchers have demonstrated that DCLK1 (Doublecortin like kinase 1) is a cancer stem cell marker in the intestinal tumour (doi: 10.3390/cancers12020274; 10.1038/ng.2481; 10.3390/cancers12123801). Moreover, Nevi et al. (doi: 10.1002/hep.31571) have demonstrated that DCLK1 could be a putative tumour marker by blood serum analysis. Based on this background, I suggest to analyse DCLK1 serum expression on patients before and after the pembrolizumab treatment, and also DCLK1 expression in tumour tissues by IHC analysis.
Furthermore, I would like to suggest to add as limitation of this study the small cohort analysed
Author Response
Response to reviewer 1
Comments
Point 1 : In my opinion, authors should add very few points in their work.
In fact, as demonstrated in many papers the biliary tree as origin by primitive gut, and show a very similar morphology, especially in the extra-hepatic tract. Many researchers have demonstrated that DCLK1 (Doublecortin like kinase 1) is a cancer stem cell marker in the intestinal tumour (doi: 10.3390/cancers12020274; 10.1038/ng.2481; 10.3390/cancers12123801). Moreover, Nevi et al. (doi: 10.1002/hep.31571) have demonstrated that DCLK1 could be a putative tumour marker by blood serum analysis. Based on this background, I suggest to analyse DCLK1 serum expression on patients before and after the pembrolizumab treatment, and also DCLK1 expression in tumour tissues by IHC analysis.
Furthermore, I would like to suggest to add as limitation of this study the small cohort analysed
Response 1 : The two points you suggested were reflected in the discussion section.
Thanks for your consideration.
Please see the attachment for details.

Reviewer 2 Report
In the present study, Chan Su Park et al. aim to identify prognostic factors associated with better treatment response to PD-1 inhibitor pembrolizumab as a second-line therapy in patients with advanced gemcitabine-refractory biliary tract cancer (BTC). Only some BTC types respond to pembrolizumab, but there are no known prognostic factors to predict its treatment benefits, therefore identification of these prognostic factors is a next step towards the development of personalized cancer therapy for patients with advanced BTC. Authors analyzed 142 patients who received PD-1 inhibitor pembrolizumab as a second-line or higher treatment. Based on their inclusion and exclusion criteria, 62 patients were excluded from the study and 80 patients were further analyzed in their study. High albumin levels and LMR and low ALP levels and TB were significantly associated with better OS in gemcitabine-refractory patients treated with pembrolizumab. The study by Chan Su Park et al. is quite interesting, and the manuscript is clear and concise. However, the following points need to be considered:
1) Page 1. Simple summary: Space is missing before (BTC).
2) Page 1. Simple summary: better use second-line, and first-line chemotherapy
3) Page 6. Legend of Figure 2. Median OS was…
Author Response
Response to Reviewer 2
Comments
Point 1 : Page 1. Simple summary: Space is missing before (BTC).
Response 1 : It's corrected
Point 2 : Page 1. Simple summary: better use second-line, and first-line chemotherapy
Response 2 : It's corrected
Point 3 : Page 6. Legend of Figure 2. Median OS was…
Response 3 : It's corrected
Thanks for your consideration
Please see the attached file for details
